# Topographic axonal projection at single-cell precision supports local retinotopy in the mouse superior colliculus

Dmitry Molotkov[1,4], Leiron Ferrarese[1,4], Tom Boissonnet [1,2,3] & Hiroki Asari [1] ✉

Retinotopy, like all long-range projections, can arise from the axons themselves or their targets. The underlying connectivity pattern, however, remains elusive at the fine scale in the mammalian brain. To address this question, we functionally mapped the spatial organization of the input axons and target neurons in the female mouse retinocollicular pathway at single-cell resolution using in vivo two-photon calcium imaging. We found a near-perfect retinotopic tiling of retinal ganglion cell axon terminals, with an average error below 30 μm or 2° of visual angle. The precision of retinotopy was relatively lower for local neurons in the superior colliculus. Subsequent data-driven modeling ascribed it to a low input convergence, on average 5.5 retinal ganglion cell inputs per postsynaptic cell in the superior colliculus. These results indicate that retinotopy arises largely from topographically precise input from presynaptic cells, rather than elaborating local circuitry to reconstruct the topography by postsynaptic cells.

Topographic organization is central to brain function[1,2]. Connections between brain regions are often spatially arranged to have one-to-one mappings, and sensory processing relies on the transmission of topographically preserved information from receptor organs. In the visual system, for example, topographic visual representations are first formed in the retina and conveyed to the brain by retinal ganglion cells (RGCs) whose axons are bundled into an optic nerve[3]. In general, neighboring RGCs project to neighboring regions in their targets[4–7]. This forms a retinotopic map in the primary retinorecipient areas, such as the lateral geniculate nucleus[8] and the superior colliculus[9,10] (SC), and retinotopy is likewise transferred throughout the entire visual pathways[6,11].

How precisely is topographic information transmitted between brain regions? Despite the substantial progress in our understanding of the connectivity patterns in simple nervous systems[12–14], only a global-level relationship is known even for the best-characterized cases in mammals, such as the retinal projection to SC[4–7]. During development, RGC axons reach their target locations based on

molecular guidance cues and activity-dependent refinement[15–17]. Long-range projections of dense axonal fibers, however, preclude a precise anatomical characterization of the connectivity patterns at single-cell resolution[18]. In the optic nerve, RGC axons are mixed and lose retinotopic organizations[19,20]. While pretarget sorting of the axons partially restores the topography in the optic tract[16,21], a fundamental question is then if RGC axons can nevertheless find precise targets even after they get lost in the long-range projection (Fig. 1, Model 1), or if their target neurons need to reconstruct the topography by elaborating local circuitry (Fig. 1, Model 2). A recent study using a high-density electrode has implicated retinotopic RGC projections to the mouse SC along the probe shank[22], favoring the former scenario. However, the precision of retinotopy at the level of individual axons and its relationship to that of target neurons remain elusive in any retinorecipient area of the mammalian nervous system.

To address this question, we performed functional mapping of RGC axon terminals and local neurons in SC at single-cell resolution in awake head-fixed mice using two-photon calcium imaging.

[1]Epigenetics and Neurobiology Unit, EMBL Rome, European Molecular Biology Laboratory, Monterotondo 00015, Italy. [2]Collaboration for joint PhD degree between EMBL and Université Grenoble Alpes, Grenoble Institut des Neurosciences, La Tronche 38700, France. [3]Present address: Center for Advanced Imaging, Heinrich-Heine-Universität Düsseldorf, Düsseldorf 40225, Germany. [4]These authors contributed equally: Dmitry Molotkov, Leiron Ferrarese. ✉e-mail: asari@embl.it

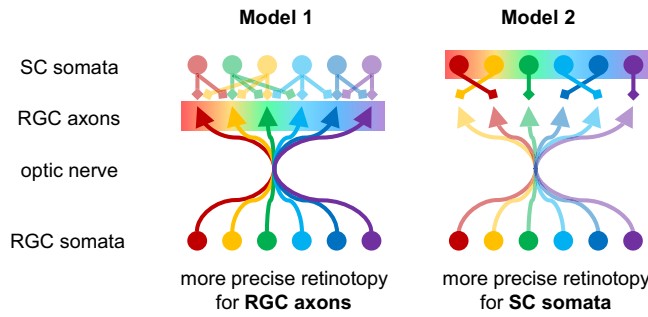

**Fig. 1 | Possible retinocollicular projection models for topographic information transmission.** Retinotopy in the superior colliculus (SC; color-coded) can arise from either precisely retinotopic projection of retinal ganglion cell (RGC) axons, innervating exact target locations in SC (Model 1); or roughly retinotopic projection of RGC axons while SC neurons identify appropriate partners to recover the retinotopy (Model 2). In Model 1, retinotopy can be more precise for RGC axons than for SC somata if the synaptic connectivity is not topographically well organized. In Model 2, in contrast, SC somata should show more precise retinotopy than RGC axons.

Subsequent analyses of their spatial organizations allowed us to quantify and directly compare the precision of retinotopy between the pre- and postsynaptic sides. Moreover, these experimental data served as a basis for a computational modeling analysis for inferring key features on the topographic information transmission in the retinocollicular pathway. Specifically, by comparing the observed and simulated retinotopy patterns, here we addressed (1) to what extent RGC axon terminals are deviated from their retinotopically optimal target locations in SC; and (2) on average how many RGCs connect to individual SC neurons.

## Results

For functional mapping of the mouse retinocollicular projections, we expressed axon-targeted calcium indicators[23] (GCaMP6s) in RGCs via intravitreal injection of recombinant adeno-associated viruses (AAVs) harboring the pan-neuronal human synapsin (hSyn) promoter and monitored the visual responses of the labeled RGC axons in SC using in vivo two-photon microscopy (Fig. 2a, b and Supplementary Movie 1). To segment axonal patches of individual RGCs and isolate their activity, we used constrained non-negative matrix factorization (CNMF) that allows us to extract morphological and temporal features from noisy time-lapse calcium activities based on their spatiotemporal correlations[24]. For a field of view of ~0.3 mm² (0.57-by-0.57 mm), we detected 26 ± 9 axonal patches displaying independent activity patterns (median ± median absolute deviation here and thereafter unless otherwise noted; n = 37 recording sites in total from 15 animals; e.g., Fig. 2c, d). The size of the individual axonal patches (135 ± 25 μm; n = 969; Supplementary Fig. 1a) is consistent with the past anatomical measurement[18], with a substantial overlap between them over the SC surface (49 ± 11%). This supports a successful signal extraction and a good coverage of RGC axonal labeling. These data allowed us to faithfully reconstruct the local two-dimensional (2D) map of the individual RGC axon terminals in SC (e.g., Fig. 2d).

The presence of a well-defined receptive field (RF) is a characteristic feature of all RGC types[3,25]. To map the RF of the identified RGC axons, we computed the response-weighted average of the presented random checkerboard stimuli (frame rate, 4 Hz; rectangular fields, 3.7° in width and 2.9° in height; e.g., Fig. 2e) and fitted a 2D Gaussian at the peak latency to characterize the spatial RF profile. Most identified axonal patches had RFs within the stimulation screen (±22° in elevation and ±36.5° in azimuth from the mouse eye[26]). In accordance with ex vivo retinal physiology[25], the average RF size of the RGC axons was 4.8 ± 1.2° (n = 719; Supplementary Fig. 1b), estimated as the

mean of the long- and short-axis diameters of the 2D Gaussian profile at 1 standard deviation (SD). There was a weak but statistically significant correlation between the RF size and the RGC axonal patch size (Pearson's r = 0.19, p = 5e-7). The RFs locally tiled the visual field with 10 ± 5 % overlap at 1 SD Gaussian profiles, where the center of every RF occupied a unique location in the visual field. This ensures that these RFs belong to different RGCs because the RF center location of RGC axons should correspond well to the location of their somata in the retina. Hence, the RF tiling faithfully represents retinotopy.

How well does the tiling pattern of RGC axons in SC agree with that of their RFs? As expected from the global retinotopy in SC[9,10,22], relative positions of the RGC axonal patches (e.g., Fig. 3a) agreed well with those of the corresponding RFs (e.g., Fig. 3b) regardless of their cell types. For quantification, we first computed the Delaunay triangulation using the geometric centers of the individual axonal patches or the RF center locations as landmark points in each space (e.g., Fig. 3c, d, respectively). This triangulation features the adjacency relationship regardless of their absolute positions, where all the adjacent pairs of the landmark points in a given space are connected as a dual graph of the Voronoi tessellation that separates the space into territories close to each landmark point. The distances between the centers of neighboring RGC axons and those between their RFs were identified to be 100 ± 30 μm (n = 761 pairs; Supplementary Fig. 1c) and 7.2 ± 2.7° (n = 776 pairs; Supplementary Fig. 1d), respectively. As a measure of the agreement between the two tiling patterns, we then calculated the fraction of the common edges between the two Delaunay triangulations (blue edges; 88.2% for the example in Fig. 3c, d). Throughout our datasets, we found a near-perfect match between the tiling patterns of RGC axons in SC and their RFs (84 ± 5%; n = 36 recordings from 12 animals; Fig. 4a) regardless of the recording depth within the superficial SC layer (120–220 μm deep from the surface; Fig. 4b). This observation is consistent with the precise axonal projection model (Model 1 in Fig. 1) whereby RGC axon terminals retinotopically tile the SC surface at single-cell precision.

To further quantify the precision of the RGC axonal projection, we compared the observed position with the ideal one that forms perfect retinotopy (e.g., Fig. 3e). Specifically, we used a linear method to estimate this retinotopically optimal tiling pattern of RGC axons: i.e., an Affine transformation that best mapped the observed RF tiling pattern onto the corresponding observed axonal tiling pattern (see "Methods" for details). We found that the average discrepancy between the observed and retinotopically optimal RGC axonal locations in SC was 27 ± 4 μm (Fig. 4c). This is much shorter than the distance between neighboring RGC axon centers (100 ± 30 μm, n = 761 pairs; Supplementary Fig. 1c) or the axonal patch size (135 ± 25 μm, n = 969; Supplementary Fig. 1a), and thus will not have a measurable impact on the retinotopy. Likewise, the extent to which the observed RF tiling pattern deviated from the linear optimal one (1.9 ± 0.3°; Fig. 4d; see Fig. 3f for example) was much smaller than the RF size of RGC axons (4.8 ± 1.2°, n = 719; Supplementary Fig. 1b) or the spacing between the RFs (7.2 ± 2.7°, n = 776 pairs; Supplementary Fig. 1d). These data support that RGCs can precisely innervate their axons to their target locations and faithfully transmit the information about retinotopy despite a loss of topographic organization along the optic nerve[19,20].

Thus far we have focused on the local retinotopy at the presynaptic input level and demonstrated a precise topographic organization of the RGC axons in the mouse SC (Figs. 2–4). What about the postsynaptic side? Taking a similar approach, we next examined the retinotopy of SC somata at single-cell resolution (Supplementary Fig. 3). Specifically, using in vivo two-photon calcium imaging, we mapped the RFs of local neurons in the superficial SC layer (58 ± 27 cells per recording from 20 animals; RF size, 5.1 ± 0.9°; RF overlap, 37 ± 11%; n = 1191 cells in total; Supplementary Fig. 1b), and performed the same tiling pattern analysis using the Delaunay triangulation

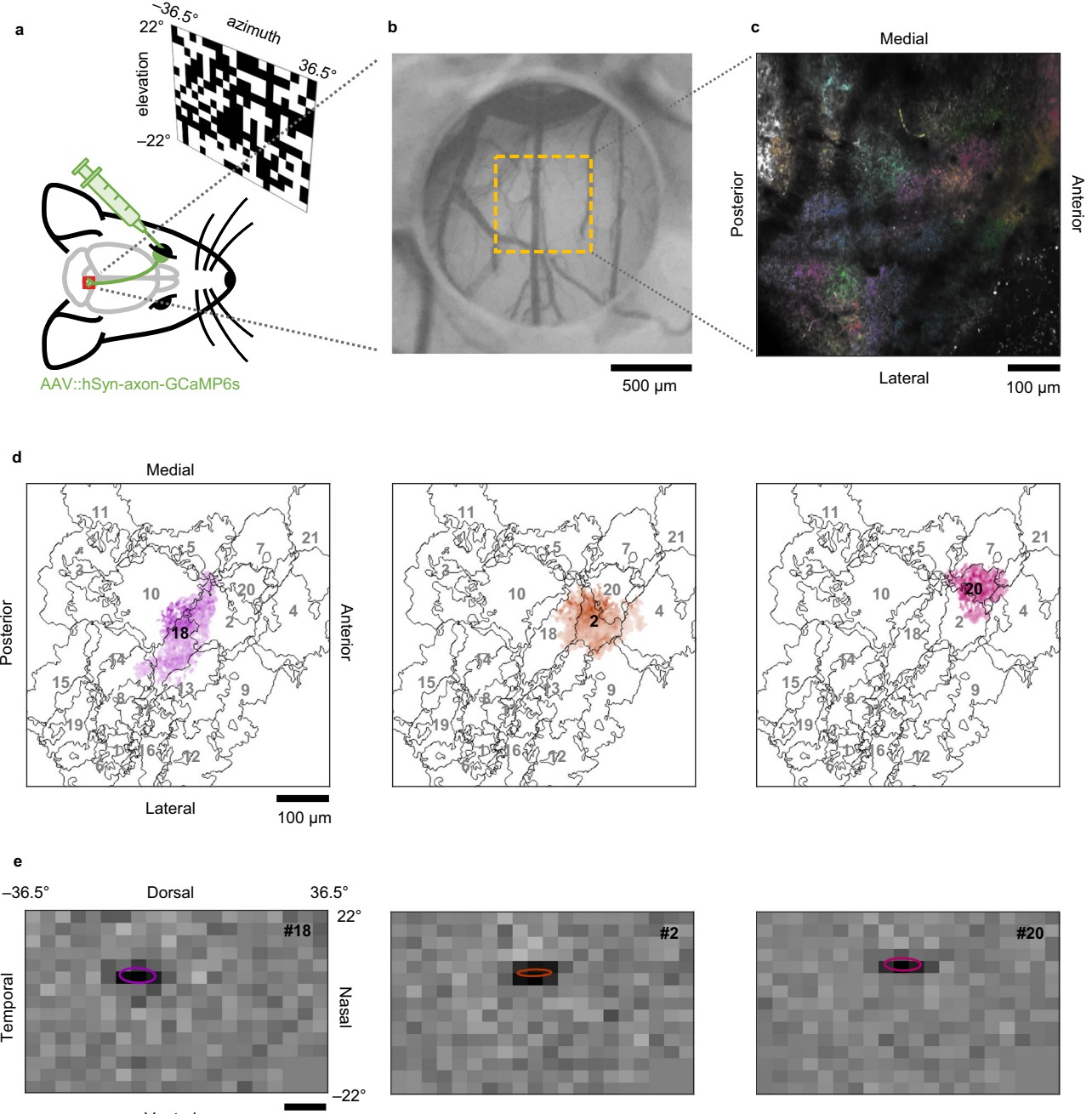

**Fig. 2 | In vivo two-photon calcium imaging of retinal ganglion cell axon terminals in the mouse superior colliculus. a** Schematic diagram of the experimental setup for retinal ganglion cell (RGC) axonal imaging in the mouse superior colliculus (SC). **b** Representative image of cranial window (*n* = 15 for axon imaging and 20 for somatic imaging). Medial-posterior part of SC was clearly visible through a cylindrical silicone plug attached to a glass coverslip. **c** Average intensity projection of representative axonal imaging data, overlaid with detected RGC axonal patches (*n* = 21; color-coded). See also Supplementary Movie 1. **d** Footprint of three representative RGC axonal patches (#18, 2, and 20 in distinct color; from left to right, respectively) overlaid with the profile of the rest patches (in gray). **e** Corresponding receptive field of the three representative RGC axonal patches (from **d**), estimated by reverse-correlation analysis (ellipse, 1 standard deviation Gaussian profile).

(neighboring somata distance, 56 ± 28 μm, *n* = 2292 pairs, Supplementary Fig. 1c; neighboring RF distance, 4.1 ± 1.9°, *n* = 2287 pairs, Supplementary Fig. 1d). As expected, we found that the tiling patterns of SC somata and their corresponding RFs agreed well in general (77 ± 5%, Fig. 4a). However, the agreement was significantly lower for SC somata than for RGC axons (*p* = 0.001, rank sum test; Fig. 4a), indicating that local cellular-level retinotopy is less precise for the postsynaptic neurons than for the input axons. Moreover, the average discrepancies between the observed and retinotopically optimal

locations of SC somata (23 ± 5 μm; Fig. 4c) or their RFs (1.8 ± 0.3°; Fig. 4d) were comparable to those for RGC axons (*p* = 0.27 and 0.66, respectively; rank sum test), suggesting that the connectivity between RGC axons and SC neurons is not necessarily made to improve the precision of local retinotopy. Thus, our data disagree with the selective connectivity model whereby SC neurons selectively integrate inputs from appropriate presynaptic partners to reconstruct the topography at single-cell resolution (Model 2 in Fig. 1). Instead, we suggest that retinotopy in SC arises primarily from precise RGC axonal projections

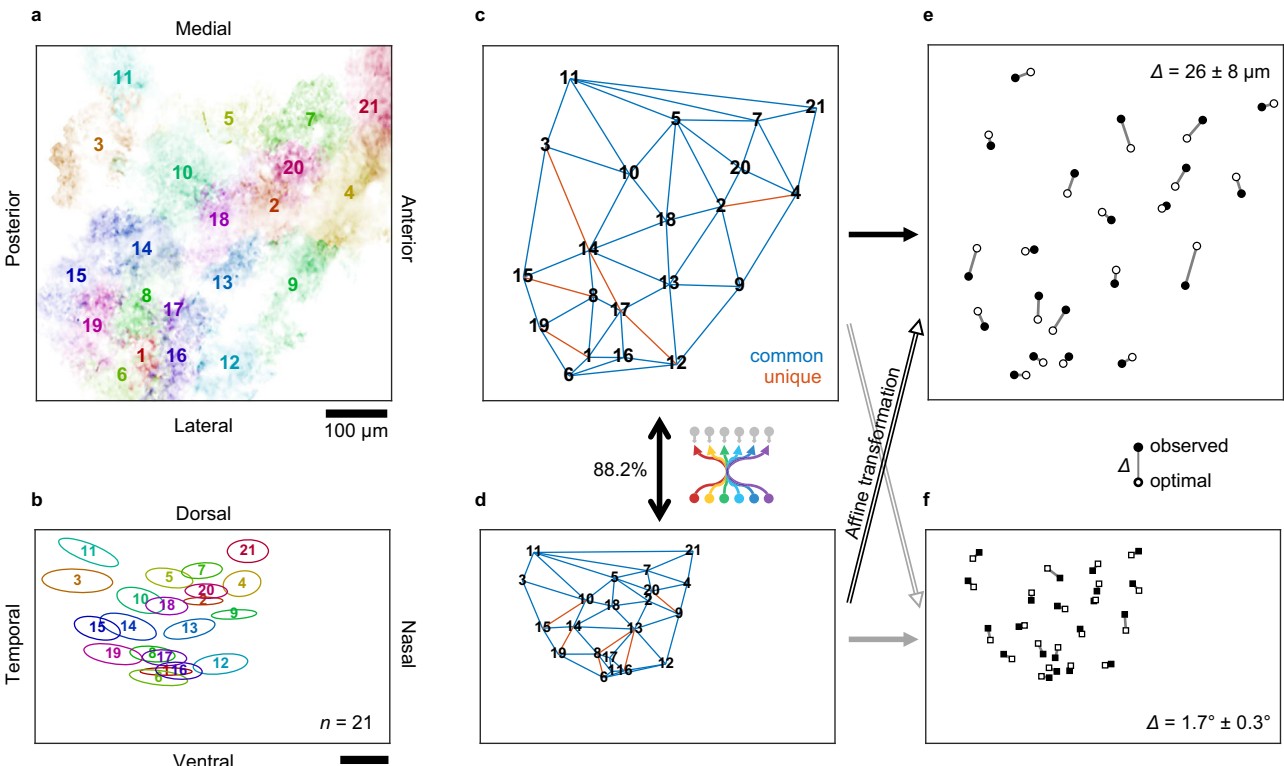

**Fig. 3 | Retinal ganglion cell axons retinotopically tile the mouse superior colliculus at single-cell precision.** See Supplementary Fig. 2 for another example. **a**, **b** Retinal ganglion cell (RGC) axon patches in the superior colliculus (SC) from a representative recording session (**a**; n = 21, color-coded; from Fig. 2c) and their corresponding receptive fields (RF; ellipse, 1 standard deviation Gaussian profile; **b**). **c**, **d** Delaunay triangulation of the RGC axonal patch locations (**c**; from **a**) and the RF centers (**d**; from **b**). The two triangulation patterns are nearly identical (88.2%; blue, common edges in both patterns; red, unique edges only in either pattern). **e** Comparison between the observed tiling pattern of RGC axons (filled circles; from **c**) and the retinotopically ideal pattern (open circle) obtained by applying an optimal Affine transformation to the corresponding RF locations (in **d**) that minimizes the discrepancy between the two patterns ($\Delta = 26 \pm 8$ μm). **f** Corresponding comparison between the observed (filled squares; from **d**) and ideal (Affine-transformed pattern in **c**) RF tiling patterns of RGC axons ($\Delta = 1.7 \pm 0.3°$).

(Model 1 in Fig. 1), without much need to elaborate the postsynaptic connectivity.

Our tiling pattern analysis showed a near-perfect retinotopy already at the level of the axonal inputs to the mouse SC and no further improvement in the precision of retinotopy for local SC neurons (Figs. 3 and 4). What are the conditions to achieve such topographic organizations in the retinocollicular pathway? To address this question, we next performed a computational modeling analysis (see "Methods" for details). Specifically, by comparing the observed and simulated tiling patterns on the pre- and postsynaptic sides, here we quantified the following two parameters: (1) the precision of RGC axonal projection to a target location in SC (Fig. 5); and (2) the number of connecting RGCs to individual SC neurons (Fig. 6). Here we did not consider any structural plasticity in our model because the focus is not on the developmental process but on the end result of the axonal organization in adult mice.

We first modeled the tiling patterns of RGC axons at different jitter levels to identify how small the projection error needs to be to recapitulate the observed precision of retinotopy (Fig. 5a). The tiling pattern of RGC somata—or equivalently, that of RGC RFs—was simulated as a 2D hexagonal lattice with a small additive Gaussian noise, where the standard deviation of the jitter followed 10% of the lattice spacing to replicate the dense packing of the cell bodies in the retina (e.g., Fig. 5b). The tiling pattern of RGC axons in SC was then simulated by introducing additional Gaussian noise to the simulated RGC RF tiling pattern, where the standard deviation σ of this additional noise determines the jitter level of the axonal projection (e.g., Fig. 5c). Here we set the axonal lattice spacing to be 100 μm based on our

experimental data (Supplementary Fig. 1c), and ran the tiling pattern analysis as we did on our experimental data to quantify the precision of retinotopy in the model. As expected, the larger the jitter was, the less precise the retinotopy was (Fig. 5d). This allowed us to determine the jitter size that agreed with the observed precision level of retinotopy (84%; Fig. 4a): i.e., $\sigma = 27 \pm 4$ μm (with 95% confidence interval). This is consistent with the average discrepancy between the observed and retinotopically optimal tiling patterns ($\Delta = 27$ μm; Fig. 4c), hence validating our modeling framework and further supporting our estimate of the precision of the RGC axonal projection.

We next modeled the retinotopy of SC somata on top of the optimal RGC projection model described above (jitter size, $\sigma = 27$ μm), using the average number of RGC inputs to SC neurons, λ, as a key model parameter (Fig. 6a). The simulated tiling pattern of SC somata (e.g., Fig. 6b) was generated in a similar way to that of RGC somata, but with a lattice spacing of 56 μm based on our experimental data (Supplementary Fig. 1c). For each simulated SC cell, the RF center location was then determined as a weighted average of the RF centers of neighboring RGC axons (Fig. 6c), where we assumed that the number of connecting RGCs followed a Poisson distribution (mean, λ), and that the connectivity strength was proportional to the amount of overlap between the SC cell's dendritic field (radius, 200 μm)[27] and the RGC's axonic field (135 μm; Supplementary Fig. 1d). Here we introduced a rather simple connectivity rule as implicated by our experimental data solely from the retinotopy viewpoint (Figs. 3 and 4), while details on the cell-type specific connectivity are beyond the scope of our modeling framework. The tiling pattern analysis on the simulated SC cells then showed that the larger the number of connecting RGCs was, the

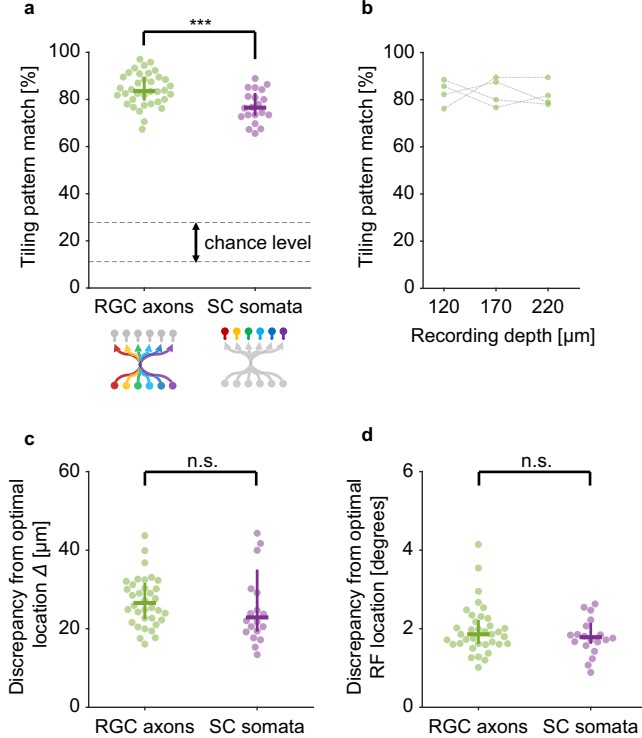

**Fig. 4 | Tiling patterns of retinal ganglion cell axons are retinotopically more precise than those of local neurons in the mouse superior colliculus. a** Retinal ganglion cell (RGC) axons (84 ± 5%; *n* = 36 recordings from 12 animals; e.g., Fig. 3) showed a significantly higher precision of retinotopy than superior colliculus (SC) somata (77 ± 5%; *n* = 20 recordings from 20 animals; e.g., Supplementary Fig. 3); *p* = 0.001, rank sum test. The tiling pattern match was defined as the number of common edges, divided by the mean of the total number of edges in the two triangulation patterns (see "Methods" for details). The error bars show the median and interquartile range; and the dotted lines represent the chance level at *p* = 0.05 (11–28%; bootstrap with 10,000 repetitions). **b** The precision of the retinotopic tiling of RGC axons was not dependent on the projection depth from the SC surface (*n* = 4 animals; 83 ± 5, 83 ± 6, and 82 ± 5% at 120, 170, and 220 μm deep, respectively; mean ± standard deviation; *p* = 0.9, one-way analysis-of-variance). **c** Deviation between the observed and retinotopically ideal locations of the RGC axons (27 ± 4 μm; e.g., Fig. 3e) or SC somata (23 ± 5 μm; e.g., Supplementary Fig. 3h); *p* = 0.27, rank sum test. The error bars show the median and interquartile range. **d** Deviation between the observed and retinotopically ideal RF locations of the RGC axons (1.9 ± 0.3°; e.g., Fig. 3f) or SC somata (1.8 ± 0.3°; e.g., Supplementary Fig. 3i); *p* = 0.66, rank sum test. The error bars show the median and interquartile range. Source data are provided as a Source Data file.

more precise the retinotopy was (Fig. 6d). This suggests that the observed relatively less precise retinotopy for SC somata (77% as opposed to 84% for RGC axons; Fig. 4a) results from a low input convergence. Indeed, by comparing the model outcome with the experimental data, here we derived that on average SC neurons receive inputs from $\lambda = 5.5 \pm 1.0$ RGCs (with 95% confidence interval). This is consistent with the observation in the previous electrophysiological studies[22,28].

## Discussion

Using in vivo two-photon axonal imaging, we conducted functional mapping of the retinocollicular projection in adult mice and demonstrated a precise retinotopic tiling of RGC axon terminals in SC at single-cell resolution (Figs. 2 and 3). Here we calculated the projection error size in two different ways: (1) based on the deviation from a linearly estimated retinotopically ideal target location (Figs. 3 and 4), and (2) by data-driven computational modeling (Fig. 5). Both methods consistently found that the projection jitter was below 30 μm (or

equivalently 2° of visual angles), much smaller than the observed RGC axonic field size (135 μm; Supplementary Fig. 1). These long-range axons can thus be innervated to their exact target locations to faithfully transmit topographic information from the retina, despite a loss of topography in the optic nerve[19,20]. Our results highlight the precision of the developmental processes, from genetically determined sorting of RGC axons[16,17] to activity-dependent structural plasticity in SC[15].

In contrast, we found that the local retinotopy of SC somata was no better than that of RGC axons (Fig. 4 and Supplementary Fig. 3). The connectivity between RGCs and SC cells is thus not necessarily made to retain or improve the topography. Instead, assuming no selectivity in the connectivity patterns, our modeling analysis indicates that a reduced precision of local retinotopy on the postsynaptic side can be a direct consequence of a low input convergence level (Fig. 6). Based on our experimental data, we derived from our model that on average SC neurons receive inputs from ~5.5 RGCs (Fig. 6). While connectivity patterns between specific RGC and SC cell types remain an open question, this is consistent with the past electrophysiological measurements[22,28], justifying our model framework and conclusion.

Taken together, we suggest that retinotopy in the mouse SC arises largely from topographically precise projection of RGC axons, rather than local circuit computation by SC neurons. While here we studied only the medial-posterior part of the superficial SC (Fig. 2), we expect that this type of organization exists across SC, given that the mechanisms underlying the retinotopy development are not dependent on the spatial location of SC[6,7]. Note, however, that the precision of axonal projection was not perfect. Postsynaptic circuit mechanisms should then be indispensable as well in retaining topography, especially for higher-order processing because otherwise, retinotopy will no longer be recognizable after a cascade of signal transmission along the visual hierarchy (e.g., below chance level after six ~80% precision transmissions). It is a future challenge to investigate the cellular-level topographic organization in other brain areas, including the retinotopy in the downstream visual pathways[11], and clarify the contribution of pre- and postsynaptic circuit mechanisms in each area.

Having a precise retinotopy at single-cell resolution facilitates spatial information processing not only at a global level but also at local circuit levels. For example, looming detection has been suggested to arise de novo in the superficial SC layer[29]. In principle, this can be achieved even in the absence of retinotopy by elaborating the wiring among local neurons. It is, however, much more efficient to exploit precise topographic information conveyed from the retina because the connectivity length and its complexity can be minimized to locally process spatial information at any point in the visual field. This will also help align different topographic maps in the same brain area to function coherently, such as the retinotopy, orientation, and ocular dominance maps in SC[30,31]. The observed precise spatial organization we demonstrate here suggests that the wiring efficiency indeed matters for local circuit computation.

How can then such a precise retinotopic projection be formed? Retinotopic map formation in SC occurs during the first postnatal week in mice, involving both genetic and activity-dependent factors[6,7]. These factors also play a key role in the development of a fine-scale organization in other sensory systems, such as the tonotopic map in the cochlear nucleus[32], and the chemotopic map in the olfactory bulb where olfactory sensory neurons expressing the same olfactory receptor type project exclusively to the same single glomerulus[33]. While overall sensory map formation is genetically predetermined by molecular cues (e.g., ephrin-Eph signaling[4,32] and axon-axon interactions[16,21]), a precise topography is established only after refinement that involves spontaneous activity, such as retinal waves during development[7,15], and eventually experience-driven alignment[5]. In particular, here we suggest that this refinement process of the retinocollicular projection during development should be extremely precise, to the extent that retinotopy arises at a single-cell resolution in adult animals (Figs. 3 and 4). It is then

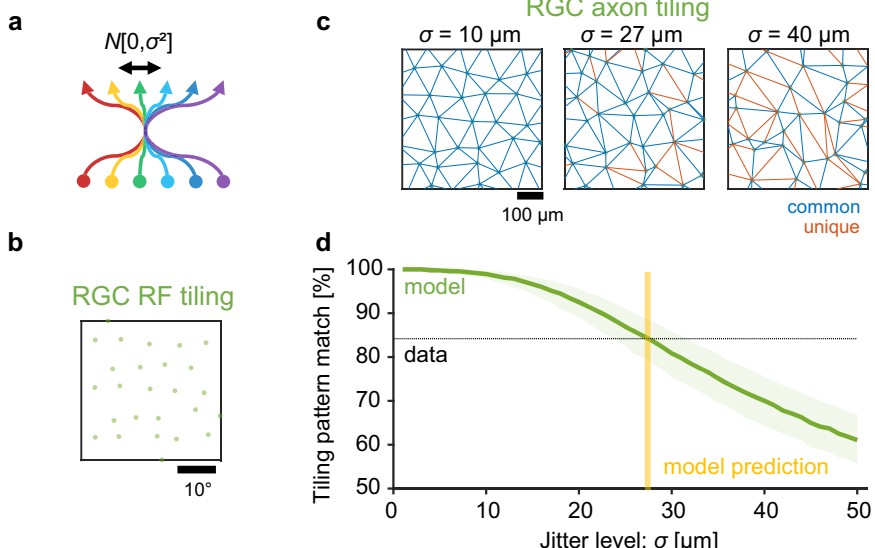

**Fig. 5 | Data-driven model prediction on the precision of retinal ganglion cell axonal projection to the mouse superior colliculus. a** Schematic of a retino-collicular projection model. We assumed that (1) a jitter of retinal ganglion cell (RGC) axonal projection follows a Gaussian distribution $N[0,\sigma^2]$; and (2) the tiling pattern of RGC receptive field (RF) centers corresponds to that of the cell locations in the retina. See "Methods" for details. **b**, **c** Representative tiling patterns of simulated RGC RF centers (**b**; on a 10%-jittered hexagonal lattice) and the corresponding axon centers at different jitter levels (c; $\sigma = 10$, 27 and 40 μm from left to right panels, respectively). Common (blue) and unique (red) triangulation edges

are also shown in each panel of (**c**) when compared to the RF tiling pattern in (**b**). The hexagonal lattice spacing was set to be 7.2° and 100 μm for RGC RFs and axons, respectively, from the experimental data (Supplementary Fig. 1). **d** Correspondence of the triangulation edges between simulated RGC RF and axon tiling patterns at different projection jitter levels (median with 95% confidence interval; 1000 repetitions). The intersection with the experimentally identified value (horizontal dotted line, 84% from Fig. 4a) gives a model prediction on the precision of RGC axonal projection (vertical yellow line; $\sigma = 27 \pm 4$ μm, with 95% confidence interval). Source data are provided as a Source Data file.

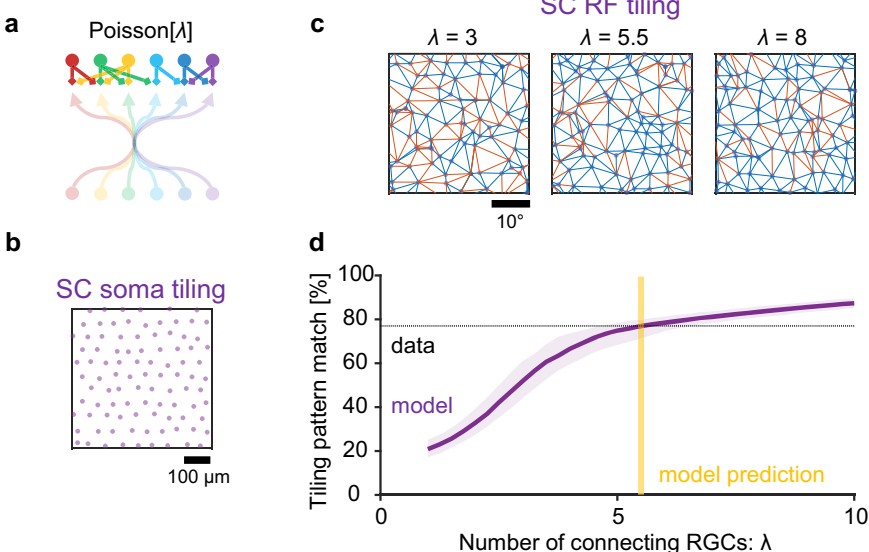

**Fig. 6 | Data-driven model prediction on the number of connecting retinal ganglion cells to individual neurons in the superior colliculus. a** Schematic of a retinocollicular mapping model. We assumed that (1) each superior colliculus (SC) neuron integrates inputs from $\lambda$ nearest neighbor retinal ganglion cell (RGC) axons, where $\lambda$ follows a Poisson distribution; and (2) the connectivity weights depend on the amount of overlap between SC dendritic field (radius, 200 μm)[27] and RGC axonic field (135 μm; Supplementary Fig. 1a). RGC axonal tiling was simulated with $\sigma = 27$ μm (Fig. 5). See "Methods" for details. **b**, **c** Representative tiling patterns of simulated SC soma centers (**b**; on a 10%-jittered hexagonal lattice with 56 μm

spacing; Supplementary Fig. 1c) and the corresponding receptive field (RF) centers at different integration levels (c; $\lambda = 3$, 5.5, and 8, from left to right panels, respectively), overlaid with common (blue) and unique (red) triangulation edges. **d** Correspondence of the triangulation edges between simulated SC soma and RF tiling patterns at different input convergence levels (median with 95% confidence interval; 1000 repetitions). The intersection with the experimentally identified value (horizontal dotted line, 77% from Fig. 4a) gives a predicted number of connecting RGCs to individual SC neurons (vertical yellow line; $\lambda = 5.5 \pm 1.0$, with 95% confidence interval). Source data are provided as a Source Data file.

possible that neuronal circuits are in general wired more precisely than previously thought to exploit topographic information for their function, including long-range projections to other retinal targets[34] as well as those in other systems, such as the callosal projections and entorhinal-hippocampal networks[1,2,35].

## Methods

No statistical method was used to predetermine the sample size. All experiments involving animals were performed under license 233/2017-PR from the Italian Ministry of Health, following protocols approved by the Institutional Animal Care and Use Committee at the European Molecular Biology Laboratory. The data analyses were done in Python 3 and Matlab R2018b-2023a (Mathworks). The statistical significance level was set to be 0.05. All summary statistics were described as median ± median absolute deviation unless otherwise noted.

### Animals

Female C57BL/6J mice (*Mus musculus*; RRID:IMSR_JAX:000664) were used at around 6–10 weeks of age at the time of the first surgery (15 for axonal imaging; 20 for somatic imaging). Mice were kept on a 12-h light/12-h dark cycle (ambient temperature, ~21 °C; humidity, 45–60%), and given water and food ad libitum. After surgery, the animals were kept in groups operated on the same day. Mice were between 12 and 24 weeks of age at the time of imaging experiments.

### Intravitreal viral injections

Intravitreal injection of recombinant adeno-associated virus (AAV) 2, pseudotyped with a hybrid of AAV1 and AAV2 capsids, was used to deliver hSyn-axon-GCaMP6s expression cassette[23] to the mouse retinal ganglion cells (RGCs). For the viral injection, mice were anesthetized (induction, 4% isoflurane in oxygen; maintenance, 1.8–2.0%) and kept on a heated plate (Supertech Physiological Temperature Controller) to avoid hypothermia. Both eyes were protected by saline drops or viscous eye ointment (VitA-POS, Ursapharm). The scleral surface on the left eye was exposed and a small piercing was made with a sterile 28-30G needle in between the sclera and the cornea. An injection pipette (~50 μm tip diameter with 30–40° bevel) prefilled with a virus solution (~1.5 × 10^14 vg mL^−1 in phosphate-buffered saline with 0.001% Pluronic F68 and 0.001% FastGreen) was then inserted into the vitreous chamber approximately 1 mm deep. The injection pipette was made from a borosilicate glass capillary (1B120F-3, WPI) with a pipette puller (DMZ, Zeitz) and a microgrinder (EG-45, Narishige). After a good sealing of the pipette was formed, 1.2 μL of the virus solution was injected at a rate of 10 nL s^−1 using a microinjection pump (either Neurostar NanoW or WPI NanoLiter 2010) with mineral oil (Sigma, M5904) filled in the displacement space by a stainless steel plunger. The pipette was slowly withdrawn at least 5 min after the completion of the injection, and the treated eye was covered with the eye ointment. The animal was then allowed to recover from anesthesia in a warmed-up chamber and brought back to its home cage.

### Intracranial viral injections

Pseudotyped AAV, composed of AAV2 *rep* and AAV9 *cap* genes or a hybrid of AAV1 and AAV2 *cap* genes, was locally injected into the mouse superior colliculus (SC) for the expression of genetically encoded calcium indicators (jGCaMP7f, jGCaMP8m or jRGECO1a) under pan-neuronal human synapsin (hSyn) promoter. The intracranial viral injection was made at the same time as the cranial implantation as described below. After making a craniotomy over the right SC, an injection pipette (~30 μm tip diameter; WPI 1B120F-3 borosilicate glass capillary pulled with Zeitz DMZ puller) prefilled with a virus solution (~5 × 10^12 to ~4 × 10^14 vg mL^−1 in phosphate-buffered saline) was inserted across the dura at coordinates from Bregma around −4 mm AP, 0.5–0.7 mm ML, and then slowly advanced until ~1.25 mm

deep. The virus solution (0.4–0.6 μL) was injected at a rate of 2 nL s^−1 with a microinjection pump (either Neurostar NanoW or WPI NanoLiter 2010). The pipette was slowly withdrawn at least 10 min after the completion of the injection, followed by the cranial window implantation procedure.

### Cranial implantations

We adapted methods described previously[30] for the cranial window implantation over the mouse SC. A cranial window assembly was made in advance, where the surface of a circular glass coverslip (5 mm diameter, 0.13–0.15 mm thickness; Assistant Karl Hecht) was activated by a laboratory corona treater (BD-20ACV Electro-Technic Products) and fused to a cylindrical silicone plug (1.5 mm diameter, 0.75–1.00 mm height; Kwik-Sil, WPI) by baking it for 24 h at 70–80 °C.

For the implantation, animals were anesthetized (induction, 4% isoflurane in oxygen; maintenance, 1.5–2.0%) and placed inside a stereotaxic apparatus (Stoelting 51625). Throughout the surgery, temperature was maintained at 37 °C using a heated plate (Supertech Physiological Temperature Controller) to avoid hypothermia, and the eyes were protected with eye ointment (VitA-POS, Ursapharm). After disinfecting and removing the scalp (Betadine 10%, Meda Pharma), the skull surface was scratched and cleaned to ensure good cement adhesion. A craniotomy of a size about 3.0 mm (anterior-posterior; AP) by 2.5 mm (medial-lateral; ML) was made over the right SC using a high-speed surgical drill (OmniDrill35, WPI) with a 0.4 mm ball-tip carbide bur (Meisinger). To prevent bleeding, the craniotomy was treated by hemostatic sponges (Cutanplast, Mascia Brunelli) soaked with sterile cortex buffer (NaCl 125 mM, KCl 5 mM, Glucose 10 mM, HEPES 10 mM, CaCl_2 2 mM, MgSO_4 2 mM, pH 7.4). For SC somata imaging, viral injections were made as described above. The implant was then placed in a way to push the transversal sinus and posterior cortex ~0.5 mm forward and position the silicone plug over the medial-caudal region of the right SC. Tissue adhesive (Vetbond, 3 M) was used to fix and seal the implant. A custom-made titanium head-plate (0.8 mm thick) was then cemented to the skull using acrylic cement powder (Paladur, Kulzer) pre-mixed with cyanoacrylate adhesive (Loctite 401, Henkel).

After the surgery, the animal was recovered from anesthesia in a warmed-up chamber and returned to its home cage. For postoperative care, animals were given intraperitoneally 5 mg kg^−1 Rimadyl (Zoetis) and 5 mg kg^−1 Baytril (Bayer) daily for 3–5 days. We waited for another 10–15 days until the cranial window completely recovered before starting in vivo two-photon imaging sessions (e.g., Fig. 2b).

### Visual stimulation

Visual stimuli were presented to the subject animals with QDSpy 0.77beta software[36]. A custom gamma-corrected digital light processing device was used to project images (1280-by-720 pixels; frame rate, 60 Hz) to a spherical screen (radius, 20 cm) placed ~20 cm to the contralateral side of an animal's eye, stimulating the visual field ±22° in elevation and ±36.5° in azimuth (Fig. 2a and Supplementary Figs. 2a and 3a). We presented (1) random water-wave stimuli (2–10 min) for generating binary masks for signal source extraction in calcium image analysis (see below); and (2) randomly flickering black-and-white checkerboard stimuli (10 min) for receptive field mapping, with rectangular fields 3.7° in width and 2.9° in height, each modulated independently by white noise at 4 Hz. When these stimuli were not presented, the screen remained uniformly gray to keep the average light intensity level constant.

### In vivo two-photon imaging

Prior to in vivo imaging sessions, animals were habituated to stay head-fixed on a custom-made treadmill disc (8–10 habituation sessions in total over a week, each for 2 h). For the imaging session, animals were kept on the treadmill with their head fixed for no longer than 2 h (2–5 sessions per animal). Two-photon calcium imaging was done on a

galvo-resonant multiphoton microscope (Scientifica HyperScope with SciScan 1.3 image acquisition software) equipped with a mode-locked tunable laser (InSight DS+, Spectra-Physics) and a plan fluorite objective (CFI75 LWD 16X W, Nikon). In each imaging session, we performed single-plane time-lapse recordings (field of view, approximately 0.65-by-0.65 mm) at a depth of 120–220 μm from the SC surface. The fluorescent signal (excitation wavelength, 920 nm for axon-GCaMP6, GCaMP7f, and GCaMP8m; 1040 nm for jRGECO1a; average laser power under the objective, 40–80 mW) was bandpass-filtered (BP 527/70 or BP 650/100 after beam-splitter FF580-FDi01, Semrock) and detected with a non-descanned gallium arsenide phosphide photomultiplier tube (Hamamatsu GaAsP PMT). Each frame was acquired with 1024-by-1024 pixels (16-bit depth) at 15.4 Hz for RGC axonal imaging, and 512-by-512 pixels (16-bit depth) at 30.9 Hz for SC somata imaging.

## Calcium image analysis

For preprocessing of RGC axon data, the original 1024-by-1024 pixel images were first downsampled to 512-by-512 pixels (2-by-2 pixels averaging) to reduce noise. To correct motion artifacts, we performed two iterations of Fourier-based rigid image registration in ImageJ (FIJI 1.51-1.54), followed by cropping the image border by 16 pixels; and then ten iterations of non-rigid motion correction (NoRMCorre) in CaImAn 1.0.1[24], followed by a 12-pixel border crop. The resulting images (456-by-456 pixels; e.g., Supplementary Movie 1) represent a field of view of around 0.57-by-0.57 mm (1.3 μm per pixel).

From the preprocessed images, we identified the axonal patches of individual RGCs and extracted their signals in CaImAn (e.g., Fig. 2c, d). Specifically, using a part of the recordings (3000–5000 frames representing the random water-wave stimulus presentation period), we first ran two iterations of constrained non-negative matrix factorization (CNMF) in CaImAn, where we set the number of expected components (*params.K*) to be 60 as an initialization parameter. From the identified components, we then manually selected those with a uniformly filled oval-like shape that had a size of around 50–150 μm as biologically relevant ones[18] and converted them into binary spatial masks to run two iterations of masked CNMF for processing the entire time-lapse recordings. The resulting set of spatial components (*estimates.A*) and deconvolved neural activities (*estimates.S*) was used for the subsequent analyses.

The area of the individual RGC axonal patches $P_i$ was estimated from the identified spatial components in CaImAn (1.3 μm per pixel), from which the radius was estimated as $(P_i/\pi)^{0.5}$ under the assumption of a circular patch shape (Supplementary Fig. 1a). The fraction of the overlap between identified axonal patches was calculated as the ratio of the areas between the intersection of any two patches $\bigcup_{i \neq j}(P_i \cap P_j)$ and the union of all patches $\bigcup P_i$.

For SC soma data, we first ran a sequence of the rigid and non-rigid motion corrections in CaImAn, followed by image cropping from 512-by-512 pixels into 480-by-480 pixels in ImageJ (1.3 μm per pixel). Using a part of the recordings (3000 frames from the random water-wave stimulus presentation period), we then ran two iterations of CNMF in CaImAn, where the images were divided into 6-by-6 (36 in total) patches and the expected number (*params.K*) and size (*params.gSig*) of neurons were set to be 5 per patch and 5-by-5 pixels in half size, respectively, as initialization parameters. From the identified putative cells, we manually selected those with a uniformly filled round shape of around 10–20 μm in size as biologically relevant ones and converted them into binary spatial masks to run two iterations of masked CNMF in CaImAn on the entire time-lapse recordings. The resulting set of spatial components (*estimates.A*) and deconvolved neural activities (*estimates.S*) was used for the subsequent analyses.

## Receptive field analysis

The receptive fields (RFs) of the identified RGC axon patches or SC somata were estimated by reverse-correlation methods using

the random checkerboard stimuli[37]. Specifically, we calculated the response-weighted average of the stimulus waveform (0.5 s window; 16.7 ms bin width) and characterized its spatial profile by the two-dimensional (2D) Gaussian curve fit at the peak latency (e.g., Fig. 2e and Supplementary Figs. 2b and 3b). The RF center was assigned to the center of that 2D Gaussian profile, and the RF size was estimated as twice the mean standard deviation (SD) of the long and short axes (Supplementary Fig. 1b). The fraction of the overlap between the RFs (1 SD Gaussian profiles) was computed similarly as for the axonal patches. The Pearson correlation coefficient between the RF size and the RGC axonal patch size was calculated with the 95% interval of the data to eliminate the outliers. Those cells that had the RF center on the border or outside the stimulus screen were eliminated from the tiling pattern analysis described below. Those recordings that had less than 10 cells with RF centers on the stimulus screen were also excluded from the tiling pattern analysis.

## Tiling data analysis

To compare the tiling patterns between RGC axon patches/SC somata and their RFs (e.g., Fig. 3 and Supplementary Figs. 2 and 3), we first computed the Delaunay triangulation of their centroid locations using the Euclidean distance in each space (e.g., Fig. 3c, d and Supplementary Figs. 2c, d and 3c, d). As a measure of similarity between the two tiling patterns, we then calculated the number of common edges, divided by the mean of the total number of edges in each triangulation. This measure is referred to as tiling pattern match in Figs. 4–6. The chance level was calculated by a bootstrap method (10,000 repetitions; Fig. 4a).

We used the least squares method to identify an optimal Affine transformation for mapping a given RF tiling pattern onto the corresponding tiling pattern of RGC axons (e.g., Fig. 3e), or vice versa (e.g., Fig. 3f). The Euclidean distance of the cell or RF locations between the observed and affine-transformed tiling patterns was then used as a measure of the precision of local retinotopy (Fig. 4c, d).

## Modeling of retinocollicular mapping

We modeled the retinocollicular mapping in four steps (Figs. 5 and 6).

1. The tiling pattern of RGC somata was simulated as a 2D hexagonal lattice with a Gaussian jitter $N[0,\sigma^2]$. The standard deviation $\sigma$ was set to be 10% of the lattice spacing $L$ to recapitulate the dense packing of the cell bodies in the retina. We assumed that the tiling pattern of RGC RFs was equivalent to the corresponding somatic tiling pattern (e.g., Fig. 5b; $L = 7.2°$ from Supplementary Fig. 1d).
2. The tiling pattern of RGC axons in SC (i.e., retinocollicular projection) was then simulated by introducing additional Gaussian noise $N[0,\sigma^2]$ to the RGC RF tiling pattern from step 1 (e.g., Fig. 5c) but with $L = 100$ μm (Supplementary Fig. 1c). When $\sigma = 0$ μm, the tiling pattern of RGC axons is identical to the somatic tiling pattern, showing perfect retinotopy (i.e., 100% tiling pattern match).
3. The tiling pattern of SC somata was simulated as a 2D hexagonal lattice ($L = 56$ μm; Supplementary Fig. 1c) with a Gaussian jitter ($\sigma = 0.1 L$) as in step 1 (e.g., Fig. 6b).
4. The RF of each SC neuron was calculated by integrating inputs from $\lambda$ nearest neighbor RGC axons, where $\lambda$ follows a Poisson distribution. Specifically, assuming that the connectivity strength depends on the amount of overlap between SC dendritic field (radius, 200 μm)[27] and RGC axonic field (135 μm; Supplementary Fig. 1a), we defined the SC RF center location as the weighted average of the RF centers of the connecting RGCs (e.g., Fig. 6c).

To identify the precision of RGC axonal projection to SC (Fig. 5), we ran the steps 1 and 2 at different jitter levels $\sigma$ (from 0 to 50 μm in steps of 1 μm; 1000 repetitions each) and calculated the similarity between the simulated tiling patterns of RGC axons and their RFs using the triangulation method as described above (Fig. 5d). We then

determined the jitter level $\sigma$ where the simulated tiling pattern similarity agreed with the experimental data (84%; Fig. 4a).

To estimate the average number of connecting RGCs to individual SC neurons (Fig. 6), we ran the steps 1–4 at different mean values $\lambda$ (from 1 to 10 RGCs in steps of 0.25; 1000 repetitions; $\sigma = 27\,\mu m$ for step 2 from Fig. 5d) and calculated the similarity between the simulated SC somatic and RF tiling patterns using the triangulation method as described above (Fig. 6d). We then determined the input convergence level $\lambda$ where the simulated tiling pattern similarity agreed with the experimental data (77%; Fig. 4a).

### Reporting summary

Further information on research design is available in the Nature Portfolio Reporting Summary linked to this article.

## Data availability

The data used in this study are available in the FIGSHARE database under accession code: 10.6084/m9.figshare.24158658 [https://doi.org/10.6084/m9.figshare.24158658]. Source data are provided with this paper.

## Code availability

The code used in this study is available on FIGSHARE along with the relevant data under accession code: 10.6084/m9.figshare.24158658 [https://doi.org/10.6084/m9.figshare.24158658].

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

## Acknowledgements

This work was supported by research grants from EMBL (H.A.); and EMBL Interdisciplinary, International and Intersectorial Postdoctoral Fellowship (EI3POD; L.F.). The EMBL Genetic and Viral Engineering Facility is acknowledged for virus production; EMBL IT Support for the provision of computer and data storage servers; and the LAR facility for taking care of animals. We thank Lin Tian (University of California Davis) for providing

axon-targeted GCaMP construct; Thomas Euler and Luke Rogerson (University of Tübingen) for sharing Python scripts to generate random water-wave stimuli, originally from Andreas Tolias lab (Baylor College of Medicine); Simone Calabrese, Ilaria Sauve, Grace Cunliffe, and Matteo Tripodi for supporting experiments; and all the Asari lab members as well as Santiago Rompani for many useful discussions.

## Author contributions

D.M. and H.A. designed the study; D.M. and L.F. performed experiments; D.M., T.B., L.F. and H.A. analyzed the results; D.M., L.F. and H.A. wrote the manuscript; and D.M. and H.A. revised the manuscript.

## Funding

## Competing interests

The authors declare no competing interests.
