## [Peer Review File · Nature Communications]

Topographic axonal projection at single-cell precision supports local retinotopy in the mouse superior colliculusReviewers' comments:

Reviewer #1 (Remarks to the Author):

The authors investigated the retinotopic mapping in the mouse superior colliculus by two-photon imaging. The manuscript only contained one schematic model (Fig 1) and one data figure (Fig 2). In Fig. 2, every panel was over-simplified with NO supporting data. Panel 2A: just one schematic illustration with no labeling images of mouse retina or SC. From 2A to 2B: again no original data. The authors really need to show some images to convince readers. Panel 2c: only one axon terminal #2. How do readers know it was one axon terminal? Any other examples? From 2C to 2D: no original images to support all labeled numbers. Similarly for all panels, there was no original viral labeling or injection, mouse retinal histology, SC histology, no mouse numbers (Were all data from just one mouse?) The manuscript needs to be seriously revised, more data added, and figures made to support the hypothesis (which by the way is really exciting).

Reviewer #2 (Remarks to the Author):

This manuscript by Molotkov et al. takes advantage of targeted expression of GCaMP in retinal ganglion cells to examine the visuotopic organization of the retinocollicular axonal map with the stated goal of determining whether retinotopy in the colliculus is derived from a postsynaptic neuronal computation or is present in the ordered inputs themselves. Using a straightforward reverse correlation RF mapping approach they conclude that the latter is closer to the reality. This is a clever and thoughtfully designed study, though it feels extremely preliminary. A number of additional experiments and analyses would be needed before I would support publication in Nature Communications.

1. For one thing there is no serious effort made to actually match the pre and postsynaptic maps, for example using different color GECIs. If that is the experimental objective as stated, it should be performed explicitly. Indeed there are several examples from aquatic species like zebrafish and *Xenopus* that come closer to meeting this goal and which are entirely ignored by the authors.

2. There was not really much population data presented. In particular I was unsure if figure 2G was a single animal or a pooling across animals, but I suspect the former. Somewhere in the paper a figure demonstrating more than the one best-case example needs to be presented in order to determine whether the typical arrangement of inputs is truly as well organized as the authors claim based on this one example.

3. There are a few technical caveats that may undermine some of the conclusions of the study. Calcium signal in axon terminals is unlikely to purely reflect the firing of the entire cell, but rather is probably closer to the activation of axonal voltage-gated calcium channels, which are known to be distributed densely at functional presynaptic terminals. Thus can we be certain that the Calcium images reflect full axonal morphology or might they reflect a subset of selective synaptic sites strengthened due to interactions with the postsynaptic partners. This could well create the illusion of well ordered inputs when what is in fact occurring is well ordered postsynaptic cells having computationally extracted a smooth map and then re-imposing this back on their inputs by only selectively strengthening those that fire together with the postsynaptic partners. While that would be very interesting, it is almost the opposite interpretation to that of the authors. Some kind of anatomical labeling would probably be needed to distinguish these possibilities.

4. Finally, the Delaunay tessellation as a method for measuring map organization seems a bit different from the exact question the authors wish to ask. Given that axon guidance mechanisms largely target axons to crudely topographic sites in the target, one could imagine that even in the most disrupted examples of $\beta 2^{-/-}$ mice the correspondence between coarse topographic maps and receptive field

positions might not be very far from 88% (and certainly not 0%) by their analysis. It would be useful to model something like that rather than to consider randomness as the “control”.

5. One more minor point is that the authors’ description of olfactory map formation as being odorant receptor-dependent somehow implying a level of activity-dependence comparable to the role played by retinal waves seems inaccurate. Activity transduction by odorant receptors and neural activity are certainly PERMISSIVE for odorant map formation, it is quite unlikely that they are instructive in the same way retinal waves are. This analogy should be toned down considerably.

Overall the writing is excellent and the ideas and techniques applied are impressive, but the analysis is somewhat anecdotal and shallow in its current form.

Reviewer #3 (Remarks to the Author):

Molitkov et al. perform calcium imaging of RGC axons that project to the posterior SC to determine the relationship between the order of the individual axonal arborization patterns and their visual receptive field organization in the adult mouse retina. Remarkably, they find that the pattern of RGC axons and that of their RFs match very well, suggesting that visual map topography is preserved even at the single cell level. The authors interpret this to mean that retinotopy arises from a precisely retinotopic projection of RGC axons (Figure 1A) rather than arising from a roughly topographic projection that is organized by SC target neurons (Figure 1B). Overall this work as presented is convincing and will likely be of interest to visual neuroscientists, however I have a few questions about the data that need to be clarified.

1. The authors are visualizing only a small part of the SC located in the superficial SC. Therefore it may not be true that this type of organization is found in all locations. This point should be emphasized in the discussion.

2. Can the authors distinguish between On, Off, and DS RGC axons? The superficial SC is rich with DS inputs and response properties and unless these cells are of the same class it is not clear why they need to tile the SC. Would ON/OFF RGCs be detected with the white noise stimulation?

3. Figure 2 data seems to come from one mouse. It would be useful to have a way to assay the reproducibility of this finding by showing the results of each mouse and the associated errors in the data.

4. Figure 2H: is there any direct evidence of where the RGC cell bodies are in the retina? I do not see how this can be determined from the data presented.

5. Line 78 “Most identified axonal patches had RFs within the stimulation screen ($\pm 22^\circ$ in elevation 79 and $\pm 36.5^\circ$ in azimuth from the mouse eye; Boissonnet et al., 2022).”

Please elaborate with numbers. How many exceptions are there? Where are these in the figure?

6. Figure 2B,D need scale bars.

7. Please show each RF as in 2C as a supplemental figure.

8. First sentence “Topographic organization is essential for the brain function (Jbabdi et al., 2013; Patel et al., 2014)”. Patel, G. H., Michael, D. K., Snyder, L. H. (2014)

I looked up these references and see no evidence that topography is essential for brain function.

We thank the reviewers for their constructive feedback (in blue below). We have made a full revision of the manuscript to extend the scope of the study and strengthen our conclusions. Below is our point-by-point reply (in black).

Reviewer #1 (Remarks to the Author):

The authors investigated the retinotopic mapping in the mouse superior colliculus by two-photon imaging. The manuscript only contained one schematic model (Fig 1) and one data figure (Fig 2). In Fig. 2, every panel was over-simplified with NO supporting data. Panel 2A: just one schematic illustration with no labeling images of mouse retina or SC. From 2A to 2B: again no original data. The authors really need to show some images to convince readers. Panel 2c: only one axon terminal #2. How do readers know it was one axon terminal? Any other examples? From 2C to 2D: no original images to support all labeled numbers. Similarly for all panels, there was no original viral labeling or injection, mouse retinal histology, SC histology, no mouse numbers (Were all data from just one mouse?) The manuscript needs to be seriously revised, more data added, and figures made to support the hypothesis (which by the way is really exciting).

We made a full revision of our manuscript to address the reviewer's concerns. In the earlier manuscript, we made 23 recording sessions from 10 mice and reported that the tiling patterns of retinal ganglion cell (RGC) axons in the superior colliculus (SC) and their receptive fields (hence somatic tiling patterns in the retina) agreed well at a single-cell resolution.

In the revised manuscript, we made substantially more recordings of RGC axons (37 recordings from 15 mice, reaching 969 RGC axonal patches in total) at different depths, as well as of local neurons in SC (1191 cells in total from 20 recordings in 20 mice). Furthermore, we performed data-driven computational modelling analyses to quantify and better characterize the retinotectal projection. These new experimental data and analyses expanded the scope of our work, and brought the following new conclusions (with an underline to highlight the new results):

1. RGCs made a near-perfect retinotopic projection to the mouse SC. This is consistent across depth in the superficial layer of SC (120-220 μm deep from the surface).
2. Each axon reached the target location with an average error below 30 μm (or equivalently, the visual angle of 2 degrees). This is much smaller than the spread of the axon terminals (135 μm), hence faithfully transmitting the topographic "input" information from the retina to SC.
3. Local SC neurons, in contrast, showed relatively less precise retinotopy.
4. Subsequent data-driven modelling ascribed it to a low input convergence level, and identified that SC neurons received inputs from around 5.5 RGCs on average.

These results indicate that retinotopy arises largely from topographically precise input from presynaptic cells, rather than elaborating local circuitry to reconstruct the topography by postsynaptic cells. Such precise axonal projection may underlie the topographic organization in other systems as well, including callosal projections and entorhinal-hippocampal networks. More generally, the results of our study suggest that neuronal circuits can be wired more precisely than often thought to exploit topographic information for their function.

With these new findings and implications, we think that our fully revised manuscript – now with 6 main figures, 2 supplementary figures, and 1 supplementary movie – will be of interest to a broader community in systems neuroscience.

Reviewer #2 (Remarks to the Author):

This manuscript by Molotkov et al. takes advantage of targeted expression of GCaMP in retinal ganglion cells to examine the visuotopic organization of the retinocollicular axonal map with the stated goal of determining whether retinotopy in the colliculus is derived from a postsynaptic neuronal computation or is present in the ordered inputs themselves. Using a straightforward reverse correlation RF mapping approach they conclude that the latter is closer to the reality.

This is a clever and thoughtfully designed study, though it feels extremely preliminary. A number of additional experiments and analyses would be needed before I would support publication in Nature Communications.

1. For one thing there is no serious effort made to actually match the pre and postsynaptic maps, for example using different color GECIs. If that is the experimental objective as stated, it should be performed explicitly. Indeed there are several examples from aquatic species like zebrafish and *Xenopus* that come closer to meeting this goal and which are entirely ignored by the authors.

In the revised manuscript, we recorded local neurons in SC as well, and found that they are generally organized in a retinotopic manner as expected from previous studies, but showed relatively less precise retinotopy than RGC axons. We further performed data-driven computational modelling analysis to characterize the retinotectal projection patterns, and identified 1) the precision of the projection to be below 30 μm (i.e., 2 degrees of visual angle); and 2) the convergence of ~ 5.5 RGCs onto SC neurons on average.

2. There was not really much population data presented. In particular I was unsure if figure 2G was a single animal or a pooling across animals, but I suspect the former. Somewhere in the paper a figure demonstrating more than the one best-case example needs to be presented in order to determine whether the typical arrangement of inputs is truly as well organized as the authors claim based on this one example.

Fig. 2G in the earlier manuscript showed the population data on the tiling pattern match between RGC axons and their receptive fields (23 recordings from 10 animals), and the example in Fig. 2A-2F showed about the average retinotopy precision as a representative, not the very best one. We apologize for the confusion.

In the revised manuscript, the representative example of RGC axon tiling pattern is shown in Fig. 3, while that of local SC cells in Supplementary Fig. 2. The population data are shown in Fig. 4 on the tiling patterns, while other statistics in Supplementary Fig. 1.

3. There are a few technical caveats that may undermine some of the conclusions of the study. Calcium signal in axon terminals is unlikely to purely reflect the firing of the entire cell, but rather is probably closer to the activation of axonal voltage-gated calcium channels, which are known to be distributed densely at functional presynaptic terminals. Thus can we be certain that the Calcium images reflect full axonal morphology or might they reflect a subset of selective synaptic sites strengthened due to interactions with the postsynaptic partners. This could well create the illusion of well ordered inputs when what is in fact occurring is well ordered postsynaptic cells having computationally extracted a smooth map and then re-imposing this back on their inputs by only selectively strengthening those that fire together with the postsynaptic partners. While that would be very interesting, it is almost the opposite

interpretation to that of the authors. Some kind of anatomical labeling would probably be needed to distinguish these possibilities.

We argue against the reviewer's point for three reasons. First, here we labelled entire RGC axons using calcium indicators fused with an axon tag (Broussard et al., 2018; see, for example, Supplementary Movie), not just the synaptic boutons with those fused with a synaptic marker (e.g., synaptophysin; Dreosti et al. 2009); and extracted the signals from the axons of the same cell as a single entity, but not separately, using a method based on a generative model (CalmAn; Giovannucci et al., 2019). Thus, local modulations at individual synapses, if any, were treated here as noise, with a minimum impact on the extracted signals. The shape and size of the extracted RGC axonal patches (Supplementary Fig. 1) are consistent with previous anatomical studies (Hong et al. 2011), supporting the successful calcium image analysis and faithful estimation of the RGC axonal tiling patterns in the mouse SC.

Second, here we mapped the receptive field (RF) of individual RGC axonal patches using reverse-correlation analysis (i.e., an average stimulus that triggered the response), which is in principle not affected by changes in the response gain (Chichilnisky, 2001). The estimated RF properties (Supplementary Fig. 1) are consistent with *ex vivo* retinal physiology (Baden et al., 2016), and the center of every RF occupied a unique location in the visual field, supporting the successful calcium image analysis. Given a good correspondence between the somatic and RF tiling patterns of RGCs, one can then safely use the obtained RF tiling patterns of RGC axons as a reference for the retinotopy analysis.

Third, we imaged local neurons in SC and found that they showed relatively less precise retinotopy than RGC axons (Fig. 4). Thus, it is reasonable to consider that information about retinotopy is transmitted primarily from presynaptic to postsynaptic sides, rather than the other way around.

4. Finally, the Delaunay tessellation as a method for measuring map organization seems a bit different from the exact question the authors wish to ask. Given that axon guidance mechanisms largely target axons to crudely topographic sites in the target, one could imagine that even in the most disrupted examples of $\beta 2^{-/-}$ mice the correspondence between coarse topographic maps and receptive field positions might not be very far from 88% (and certainly not 0%) by their analysis. It would be useful to model something like that rather than to consider randomness as the "control".

In the revised manuscript, we further quantified the precision of retinotopy for RGC axonal tiling patterns in two ways; one by comparing the observed and linear-optimal tiling patterns (e.g., Fig. 3E,F), and the other by computational modelling (Fig. 5). Both analyses obtained a consistent result that RGC axons reached the target location with an average error below 30 μm (or 2 degrees of visual angle). This is much smaller than the spatial extent of the RGC axons ($\sim 135 \mu\text{m}$), highlighting the precision of the retinotectal projection.

5. One more minor point is that the authors' description of olfactory map formation as being odorant receptor-dependent somehow implying a level of activity-dependence comparable to the role played by retinal waves seems inaccurate. Activity transduction by odorant receptors and neural activity are certainly PERMISSIVE for odorant map formation, it is quite unlikely that they are instructive in the same way retinal waves are. This analogy should be toned down considerably.

Overall the writing is excellent and the ideas and techniques applied are impressive, but the analysis is somewhat anecdotal and shallow in its current form.

We revised the discussion section to clarify that both molecular and activity-dependent cues are generally important to form topographic organization in the brain, and mentioned more examples (tonotopy in the auditory system) besides the olfactory system.

Reviewer #3 (Remarks to the Author):

Molotkov et al. perform calcium imaging of RGC axons that project to the posterior SC to determine the relationship between the order of the individual axonal arborization patterns and their visual receptive field organization in the adult mouse retina. Remarkably, they find that the pattern of RGC axons and that of their RFs match very well, suggesting that visual map topography is preserved even at the single cell level. The authors interpret this to mean that retinotopy arises from a precisely retinotopic projection of RGC axons (Figure 1A) rather than arising from a roughly topographic projection that is organized by SC target neurons (Figure 1B). Overall this work as presented is convincing and will likely be of interest to visual neuroscientists, however I have a few questions about the data that need to be clarified.

1. The authors are visualizing only a small part of the SC located in the superficial SC. Therefore it may not be true that this type of organization is found in all locations. This point should be emphasized in the discussion.

We mentioned the caveat in the revised manuscript as pointed out by the reviewer. It is, however, reasonable to expect that our finding will hold across all locations in SC, given that the retinotopy globally exists across SC and its formation is governed by the same set of molecular and activity-dependent mechanisms regardless of the location.

2. Can the authors distinguish between On, Off, and DS RGC axons? The superficial SC is rich with DS inputs and response properties and unless these cells are of the same class it is not clear why they need to tile the SC. Would ON/OFF RGCs be detected with the white noise stimulation?

White noise analysis provides only the average properties of the stimulus that triggered the responses, and cannot clearly separate the cell types by themselves, such as direction-selective RGCs. While it is not clear why RGCs need to tile SC in a retinotopic manner, our results provide stronger evidence that the retinotopic tiling holds irrespective of cell-type classification in the superficial layer of the mouse SC across depth (120-220 μm).

3. Figure 2 data seems to come from one mouse. It would be useful to have a way to assay the reproducibility of this finding by showing the results of each mouse and the associated errors in the data.

While Fig. 2A-2F in the earlier manuscript showed a representative example from one recording session, Fig 2G showed population data (23 recording sessions from 10 mice), demonstrating that the retinotopy holds at a single-cell resolution for RGC axons projected to the mouse SC.

For clarification, we included more figure panels in the revised manuscript to show the individual examples and population data separately. In particular, Fig. 2 showed the technical details of RGC axonal imaging and analyses, Fig.3 showed the representative example of the

tiling pattern analysis for RGC axons (while Supplementary Fig. 2 for local neurons in SC), and Fig. 4 showed the population data. Consistent results from even larger data sets (37 recordings from 15 mice, reaching 969 RGC axonal patches in total) at different depth (120-220 μm deep from the SC surface) support the reproducibility of our results.

4. Figure 2H: is there any direct evidence of where the RGC cell bodies are in the retina? I do not see how this can be determined from the data presented.

Due to dense labelling and long-range projections of RGC axons, we could not trace and identify the cell bodies of the RGC axons we observed. This is exactly why retinotopy has not been anatomically examined at a single-cell resolution, and why functional mapping is required for such analyses, where one can exploit the fact that the tiling of RGC receptive fields corresponds well to that of RGC cell bodies in the retina.

5. Line 78 “Most identified axonal patches had RFs within the stimulation screen ($\pm 22^\circ$ in elevation 79 and $\pm 36.5^\circ$ in azimuth from the mouse eye; Boissonnet et al., 2022).”

Please elaborate with numbers. How many exceptions are there? Where are these in the figure?

Out of $N=969$ RGC axonal patches we identified from 37 recordings made in 15 mice, we found $N=719$ with clear RFs. We included Supplementary Fig. 1 to show the probability distributions of the axonic and receptive field sizes as well as the distance between the neighboring axons/receptive fields of our data.

6. Figure 2B,D need scale bars.

We included scale bars in the revised manuscript.

7. Please show each RF as in 2C as a supplemental figure.

We included more examples in the revised manuscript. Other examples can be retrieved via shared data.

8. First sentence “Topographic organization is essential for the brain function (Jbabdi et al., 2013; Patel et al., 10 2014)”. Patel, G. H., Michael, D. K., Snyder, L. H. (2014)

I looked up these references and see no evidence that topography is essential for brain function.

We totally agree with the reviewer that, to our knowledge, there is no evidence that topography is “necessary” for brain function. However, it exists in many systems, including visual/auditory/olfactory systems, and thus one can argue that it is “central” for brain function as described in the references. We revised the sentence accordingly in the revision.

REVIEWER COMMENTS

Reviewer #1 (Remarks to the Author):

This manuscript is completely rewritten with new figures and data. It is a very exciting study with imaging data and modeling. Three points require further attention: (1) Michael Crair had a JCN paper in 2005 which should be referenced in Introduction. They had used anatomical tracing to show that RGC axons re-established dorsoventral but not Nasotemporal map after chiasm. Of course the current study used much more advanced techniques, but the idea has been there for a while. (2) Cell type specific projections of RGCs to SC needs to be carefully discussed. They may go to different regions, different layers etc. so regional changes may not be extended to the whole SC. And (3) Inayat et Al 2015 J Neuroscience showed that the superficial SC neurons are highly direction selective. type-specific SC neurons also need to be discussed.

Reviewer #2 (Remarks to the Author):

This revised manuscript by Molotkov et al uses calcium imaging in the superior colliculus to map out the topographic organization of RGC inputs and postsynaptic neurons in the adult mouse. The main finding is that axonal arbor organization in the SC is surprisingly retinotopically precise, comparable to if not exceeding that of the postsynaptic map. Based on these observations the authors propose that topographic information in the SC derives mainly from the precise organization of the inputs rather than postsynaptic processing. Instead, postsynaptic cells are more likely tasked with representing and extracting higher order properties multiplexed on the background of topographic precision.

This revised submission is vastly improved over the original manuscript and now provides enough original data to better assess the findings. The data appears to be of high quality and the analysis offers important new insight.

1. My remaining concern now lies simply with how the data are presented in a mechanistic context, without any but the most superficial consideration or discussion of developmental refinement of inputs. That is to say that the model espoused fails to acknowledge the likely possibility that an initially imprecise set of retinal inputs may have undergone a period of activity-dependent sorting involving the addition and elimination of many branches of the axon arbor (and including the participation of postsynaptic cells) as well as migration of arbors within the target, as has been well understood to take place for many decades now. I would strongly urge the authors to restate their conclusions (e.g., line 356) by more fully acknowledging that the axonal organization in the adult animals is likely the consequence of early structural plasticity events. Too much of the paper is pushing the idea that axons target the SC with near-perfect retinotopic precision (despite disorganization in the optic nerve) without accepting that this precision is likely the result of structural plasticity earlier in development.

2. This tendency to overgeneralize the adult mouse observations comes across right in the first sentence, which makes the blanket statement that fine scale connectivity underlying retinotopy remains unknown. This strikes me as way too broad a dismissal/generalization. I suspect fine-scale connectivity of retinotopy in *Drosophila*, for example, may be quite well understood. In zebrafish, a great deal of single-cell work has been performed at a level of resolution and completeness that rivals anything in mammals (e.g., Kunst, M... Baier, H. (2019). A Cellular-Resolution Atlas of the Larval Zebrafish Brain. *Neuron*, 103(1), 21–38.e5. <https://doi.org/10.1016/j.neuron.2019.04.034>). I recommend such generalizations be narrowed explicitly to mammalian brain.

3. The authors repeatedly cite work by Horton and by Guillery on disorder of RGC axons in the optic nerve ("despite a loss of topographic organization along the optic nerve (Horton et al., 1979; Colello

and Guillery, 1998). However this statement entirely ignores the fact that there is evidence of considerable pretarget sorting in the optic tract. (e.g., Pretarget sorting of retinocollicular axons in the mouse. Plas DT, Lopez JE, Crair MC. *J Comp Neurol*. 2005 Oct 31;491(4):305-19. doi: 10.1002/cne.20694 and Axon-Axon Interactions Regulate Topographic Optic Tract Sorting via CYFIP2-Dependent WAVE Complex Function. Cioni JM, Wong HH, Bressan D, Kodama L, Harris WA, Holt CE. *Neuron*. 2018 Mar 7;97(5):1078-1093.e6. doi: 10.1016/j.neuron.2018.01.027.)

4. The main analysis that forms the basis for claiming that postsynaptic retinotopy is less organized than presynaptic is the measurements of something called "tiling pattern match %". However this actual measurement is not defined anywhere in the paper, nor is it entirely self-explanatory.

Reviewer #3 (Remarks to the Author):

Molotkov et al. perform calcium imaging of RGC axons that project to the posterior SC and SC neurons at different depths to determine the relationship between the order of the individual axonal arborization patterns/SC soma locations and their visual receptive field organization in the adult mouse. Remarkably, they find that the pattern of RGC axons and that of their RFs match very well, suggesting that visual map topography is preserved even at the single axon level. The authors interpret this to mean that retinotopy arises from a precisely retinotopic projection of RGC axons (Figure 1A) rather than arising from a roughly topographic projection that is organized by SC target neurons (Figure 1B). Overall this resubmission is improved and will be an important set of experiments for the field to ponder.

Minor points:

Figure 2E and 4C: Please show the sigmoid that demarcates the RF on the white noise stimuli. Also instead of Temporal-Nasal the figure should be labeled in degree (-36 to 36?).

Multiple examples of RFs in 4C should be shown.

Figure 2 and Figure 3 are the same mouse. It would be nice to see what the data looks like for another mouse.

Video seems useless without explanation. When does the light turn on? I can not see any domain specific changes by eye.

We thank the reviewers for their constructive comments (in blue). Below are our point-by-point replies (in black).

Reviewer #1 (Remarks to the Author):

This manuscript is completely rewritten with new figures and data. It is a very exciting study with imaging data and modeling. Three points require further attention:

(1) Michael Crair had a JCN paper in 2005 which should be referenced in Introduction. They had used anatomical tracing to show that RGC axons re-established dorsoventral but not Nasotemporal map after chiasm. Of course the current study used much more advanced techniques, but the idea has been there for a while.

We are grateful for this informative remark. The suggested paper (Plas et al., 2005) is indeed highly relevant to our study. In the revised manuscript, we made relevant editing of the introduction and cited Plas et al. (2005) where appropriate.

(2) Cell type specific projections of RGCs to SC needs to be carefully discussed. They may go to different regions, different layers etc. so regional changes may not be extended to the whole SC.

We agree that the projections of different retinal ganglion cell (RGC) types require special attention as they each have a specific projection pattern (Dhande and Huberman, 2014). Here we have three reasons why we think that our conclusion will likely hold across the superior colliculus (SC). First, here we found precise tiling patterns of RGC axons in SC without discriminating their cell types (Figs. 3 and 4A). Any subset of these axons should then be tiled precisely, such as those of a given RGC type. Second, we found consistent results in our measurement across depths (120, 170 and 220 μm from the SC surface; Fig. 4B). Lastly, to the best of our knowledge, axons of each RGC type cover entirely – but not always uniformly – their target stratification layer of SC (Triplett, et al., 2014; Marterstec, et al., 2017), instead of converging to a specific region. It is still an open question whether any bias in the axonal density of a given RGC type in its target SC layer is due to the projection bias, or simply arises from the distribution bias of the somata in the retina (Bleckert, et al., 2014). Nevertheless, we think that the latter scenario is more likely, given the presence of a global retinotopy in SC.

With that said, we feel that this topic needs to be clarified in a separate study, rather than making a speculation within the present study. We would also like to point out that very recent studies (Liang et al., 2023; Malmazet et al., 2023) partially address the connectivity patterns between specific RGC and SC cell types with respect to orientation- and direction-selectivity. We are also currently working to figure out the clustering of OS/DS RGC axon terminals and its functional translation to SC cells.

And (3) Inayat et Al 2015 J Neuroscience showed that the superficial SC neurons are highly direction selective. type-specific SC neurons also need to be discussed.

We cannot deny the possibility that the retinotopy of a given SC cell type may be highly precise, while that of SC cells altogether was not as we report here (Fig. 4A and Supplemental Fig. 3). It should be noted, though, that there is no consensus yet on cell types in SC. On the one hand, superficial SC neurons can be divided into 4 types based on morphological,

electrophysiological, and genetic criteria (Gale & Murphy, 2014). On the other hand, functional classification based on visual response properties identified 24 types (Li & Meister, 2023). Moreover, a recent study indicates that SC response properties can be dynamic depending on the visual contexts (Liang et al., 2023), making it even more difficult to define a cell-type in SC. It would be interesting if identifying connectivity patterns to specific RGC types may help define SC cell types in future studies.

Reviewer #2 (Remarks to the Author):

This revised manuscript by Molotkov et al uses calcium imaging in the superior colliculus to map out the topographic organization of RGC inputs and postsynaptic neurons in the adult mouse.

The main finding is that axonal arbor organization in the SC is surprisingly retinotopically precise, comparable to if not exceeding that of the postsynaptic map. Based on these observations the authors propose that topographic information in the SC derives mainly from the precise organization of the inputs rather than postsynaptic processing. Instead, postsynaptic cells are more likely tasked with representing and extracting higher order properties multiplexed on the background of topographic precision.

This revised submission is vastly improved over the original manuscript and now provides enough original data to better assess the findings. The data appears to be of high quality and the analysis offers important new insight.

1. My remaining concern now lies simply with how the data are presented in a mechanistic context, without any but the most superficial consideration or discussion of developmental refinement of inputs. That is to say that the model espoused fails to acknowledge the likely possibility that an initially imprecise set of retinal inputs may have undergone a period of activity-dependent sorting involving the addition and elimination of many branches of the axon arbor (and including the participation of postsynaptic cells) as well as migration of arbors within the target, as has been well understood to take place for many decades now. I would strongly urge the authors to restate their conclusions (e.g., line 356) by more fully acknowledging that the axonal organization in the adult animals is likely the consequence of early structural plasticity events. Too much of the paper is pushing the idea that axons target the SC with near-perfect retinotopic precision (despite disorganization in the optic nerve) without accepting that this precision is likely the result of structural plasticity earlier in development.

We fully agree with the reviewer that the activity-dependent refinement of RGC axonal arbors during development is critical to form the precise retinotopic organization we observed in this study. This is exactly what we meant by the sentence in line 356 of the original manuscript: “this refinement process [...] should be extremely precise, to the extent that retinotopy arises at single-cell resolution [...].” For clarification, we added a phrase “during development” here in the revised manuscript. We also revised the abstract, introduction and the discussion of the manuscript accordingly, and included more references where appropriate, to acknowledge the importance of early structural plasticity (as well as the genetic factors) during development.

Note, however, that the focus of our study is not on the developmental process but on the end result of the axonal organization in adult mice. This is why we did not consider any structural plasticity in our current model, and simply asked the relationship between the tiling pattern precision and the projection error level (Fig. 5) and how the underlying connectivity pattern between RGCs and SCs should look given the observed tiling patterns (Fig. 6). The revised

manuscript clarified this point when we first described our model in the results section. It is a future challenge to incorporate the developmental process in the model.

2. This tendency to overgeneralize the adult mouse observations comes across right in the first sentence, which makes the blanket statement that fine scale connectivity underlying retinotopy remains unknown. This strikes me as way too broad a dismissal/generalization. I suspect fine-scale connectivity of retinotopy in *Drosophila*, for example, may be quite well understood. In zebrafish, a great deal of single-cell work has been performed at a level of resolution and completeness that rivals anything in mammals (e.g., Kunst, M... Baier, H. (2019). A Cellular-Resolution Atlas of the Larval Zebrafish Brain. *Neuron*, 103(1), 21–38.e5. <https://doi.org/10.1016/j.neuron.2019.04.034>). I recommend such generalizations be narrowed explicitly to mammalian brain.

We thank the reviewer for this insightful remark. We revised the manuscript, including the abstract, to address the concern on the unnecessary generalization.

3. The authors repeatedly cite work by Horton and by Guillery on disorder of RGC axons in the optic nerve (“despite a loss of topographic organization along the optic nerve (Horton et al., 1979; Colello and Guillery, 1998). However this statement entirely ignores the fact that there is evidence of considerable pretarget sorting in the optic tract. (e.g., Pretarget sorting of retinocollicular axons in the mouse. Plas DT, Lopez JE, Crair MC.J *Comp Neurol*. 2005 Oct 31;491(4):305-19. doi: 10.1002/cne.20694 and Axon-Axon Interactions Regulate Topographic Optic Tract Sorting via CYFIP2-Dependent WAVE Complex Function. Cioni JM, Wong HH, Bressan D, Kodama L, Harris WA, Holt CE. *Neuron*. 2018 Mar 7;97(5):1078-1093.e6. doi: 10.1016/j.neuron.2018.01.027.)

We thank the Reviewers #2 and #1 for bringing our attention to these important developmental processes. Following the reviewers’ suggestions, we revised the introduction and discussion sections.

4. The main analysis that forms the basis for claiming that postsynaptic retinotopy is less organized than presynaptic is the measurements of something called “tiling pattern match %”. However this actual measurement is not defined anywhere in the paper, nor is it entirely self-explanatory.

We apologize for the confusion. By “tiling pattern match,” we meant the fraction of common edges between the two tiling patterns. We fully described the “tiling data analysis” in the method section, but forgot to explicitly define the term in the original manuscript. We corrected the issue in the revised manuscript.

Reviewer #3 (Remarks to the Author):

Molotkov et al. perform calcium imaging of RGC axons that project to the posterior SC and SC neurons at different depths to determine the relationship between the order of the individual axonal arborization patterns/SC soma locations and their visual receptive field organization in the adult mouse. Remarkably, they find that the pattern of RGC axons and that of their RFs match very well, suggesting that visual map topography is preserved even at the single axon level. The authors interpret this to mean that retinotopy arises from a precisely retinotopic projection of RGC axons (Figure 1A) rather than arising from a roughly topographic projection that is organized by SC target neurons (Figure 1B). Overall this resubmission is improved and will be an important set of experiments for the field to ponder.

Minor points:

Figure 2E and 4C: Please show the sigmoid that demarcates the RF on the white noise stimuli. Also instead of Temporal-Nasal the figure should be labeled in degree (-36 to 36?).

We assume that this remark concerns Figure 2E and Supplemental Figure 2C in the original manuscript, but not Figure 4C as it does not contain the receptive field (RF) of any cell. We revised these figures following the reviewer's requests. Note that Supplemental Figure 2 in the original manuscript became Supplemental Figure 3 in the revised manuscript.

Multiple examples of RFs in 4C should be shown.

As mentioned above, we assumed that the reviewer referred to Supplemental Figure 2C in the original manuscript, and revised the figure accordingly.

Figure 2 and Figure 3 are the same mouse. It would be nice to see what the data looks like for another mouse.

As is requested, we added a supplementary figure to the revised manuscript ("Supplemental Figure 2") to show another example of RGC axon tiling in SC.

Video seems useless without explanation. When does the light turn on? I can not see any domain specific changes by eye.

The video in the original manuscript (Supplemental Movie 1) showed the representative data in the middle of checkerboard stimulation. We replaced it with the one showing data at the transition of the stimulus presentation in the revised manuscript. To clarify the timing of the stimulus onset, we overlaid a black square at the bottom-right corner of the video that turned white when the stimulus was presented.

REVIEWERS' COMMENTS

Reviewer #1 (Remarks to the Author):

The authors addressed my concerns.

Reviewer #2 (Remarks to the Author):

I have read the revised manuscript and the rebuttal of reviewers' comments and believe that all my concerns in the last draft of the paper have been addressed to my satisfaction. I would still prefer for the definition of the key measure "tiling pattern match %" to be made in the figure legend rather than in the methods, but this is up to the authors.

Reviewer #3 (Remarks to the Author):

The reviewers have answered each of my critiques sufficiently.

We thank the reviewers for their time and comments (in blue). Below are our point-by-point replies (in black).

Reviewer #1 (Remarks to the Author):

The authors addressed my concerns.

Thanks.

Reviewer #2 (Remarks to the Author):

I have read the revised manuscript and the rebuttal of reviewers' comments and believe that all my concerns in the last draft of the paper have been addressed to my satisfaction. I would still prefer for the definition of the key measure "tiling pattern match %" to be made in the figure legend rather than in the methods, but this is up to the authors.

We included the definition of the "tiling pattern match" in the legend of Figure 4.

Reviewer #3 (Remarks to the Author):

The reviewers have answered each of my critiques sufficiently.

Thanks.